# A Task-Aware Dynamic Expansion Network for Continual Reinforcement Learning

## Abstract

Reinforcement learning has been widely applied in domains such as gaming and robotic control. However, CRL methods that rely on a single network architecture often struggle to preserve previously learned skills when they are trained on substantially different new tasks. To address this challenge, we propose a Task-Aware Dynamic Expansion Network (TADEN), which features a task-aware expansion strategy. This approach collects sequential environment states to measure task similarity, which reflects the suitability of the existing policy to a new task. Then, the task similarity score is utilized to determine whether to expand the actor-critic architecture or reuse existing modules. When expanding the network, our method leverages prior knowledge while preserving adaptability by initializing new modules through the reuse of lower layers of existing modules. We evaluate our method on the MiniHack and Atari environments. The experimental results demonstrated that TADEN achieved significantly better performance and mitigated catastrophic forgetting compared to existing methods.

## 1 Introduction

Reinforcement learning (RL) has been widely applied in gaming (Sieusahai & Guzdial, 2021; Ye et al., 2021) and robotic control (Salvato et al., 2021; Cheng et al., 2023), automatic vehicle (Yan et al., 2022; Sierra-Garcia & Santos, 2024), and has achieved promising results. However, when trained under a continual learning setting, where different tasks come in a sequential manner, most RL methods suffer from catastrophic forgetting (Wang et al., 2024), losing previously acquired knowledge after learning multiple tasks (Bang et al., 2022). Addressing this challenge is essential for enabling RL agents to operate effectively in dynamic real-world environments.

In recent years, a growing number of continual reinforcement learning (CRL) (Abel et al., 2023; Muppidi et al., 2024; Kessler et al., 2023) methods have been proposed. In CRL, the agent sequentially learns multiple RL tasks to acquire distinct task-specific skills. As the number of tasks increases and task interference becomes more severe (Kessler et al., 2022), architecture-based methods exhibit superior performance (Malagon et al., 2024; Rusu et al., 2016; Ahn et al., 2025; Powers et al., 2022a; Schwarz et al., 2018; Gaya et al., 2023). These methods typically mitigate catastrophic forgetting by expanding the network capacity. Some of them (Malagon et al., 2024; Rusu et al., 2016) utilize task labels during both training and testing to identify task boundaries, which helps to select a suitable network module for different tasks to avoid forgetting. However, in real-world environments that are continually changing, explicit task labels are often unavailable, particularly during testing. Therefore, some of them (Ahn et al., 2025; Powers et al., 2022a; Schwarz et al., 2018; Gaya et al., 2023) only rely on task boundaries to incrementally expand the network during training and to perform task inference at testing time, enabling the selection of appropriate modules without explicit task labels. However, as the number of tasks increases, unbounded expansion of the network incurs high computational and memory costs. Additionally, the isolation of task-specific policies can hinder knowledge sharing and may lead to training collapse on complex tasks.

To overcome these challenges, we propose a *task-aware dynamic expansion network* (TADEN) training framework that alleviates catastrophic forgetting in CRL settings. Our proposed TADEN leverages task boundary information during training to determine whether network expansion is necessary, while eliminating the need for explicit task labels at testing time by task inference to select the optimal policy. Specifically, we propose a *task-aware expansion strategy*, which utilizes an RL

method to collect task-specific features. Then the task-specific information is stored in a memory bank, which is dynamically expanded as new tasks are encountered. With this memory bank, inter-task similarity can be effectively measured. This similarity is then used to determine whether to expand the model or reuse an existing actor–critic module, while minimizing computational overhead. By making more accurate expansion decisions, conflicting tasks can be more accurately assigned to different policies, allowing catastrophic forgetting to be better avoided. During module expansion, we propose a *dual-mode initialization strategy*. The low-level feature extraction layers of the new module are initialized from an existing module, while the top-level policy head is randomly initialized. This design promotes the effective utilization of the model's existing knowledge while maintaining its plasticity for new tasks. During the testing process, by comparing the collected states information of the testing task with existing ones in the memory bank, the most suitable sub-network can be automatically selected to perform the testing task without a task label.

We evaluated our TADEN training framework on two widely used RL environments, MiniHack and Atari game environments, and achieved substantially better average performance compared to standard CRL baselines.

## 2 RELATED WORK

CRL algorithms are designed to enable agents to learn sequentially from a stream of tasks, mitigate catastrophic forgetting, and facilitate knowledge transfer to future tasks (Powers et al., 2022b). In recent years, numerous approaches have been proposed to address the catastrophic forgetting in CRL, which can be broadly categorized into *regularization-based*, *replay-based*, and *architecture-based* methods (Meng et al., 2025). The *regularization-based* methods mitigate catastrophic forgetting by employing regularization techniques. For example, Elastic Weight Consolidation (EWC) (Kirkpatrick et al., 2017) and Online EWC (Huszár, 2018) constrain parameter updates to protect the knowledge acquired from previously learned tasks. The *replay-based* methods have been widely adopted in CRL by leveraging experience replay techniques. For example, storing data from previous tasks and jointly training with new task data (Rolnick et al., 2019; Oh et al., 2022) are utilized to consolidate existing knowledge. Furthermore, a generator network is incorporated to synthesize data (Atkinson et al., 2021; Li et al., 2021) to mitigate the privacy risks associated with storing raw samples and enable continual learning without direct access to original training data. The *architecture-based* methods have been increasingly adopted in CRL by dynamically adding network modules according to task sequences. A small task-specific network modules are added during training for each new task and later distilled into a unified backbone network to consolidate knowledge (Schwarz et al., 2018). The network expansion decisions for each RL task and task inference are made based on the estimated task value (Powers et al., 2022a; Gaya et al., 2023). The network is expanded for each RL task, and knowledge transfer is achieved by reusing policies from previous tasks (Malagon et al., 2024), and the task labels are required for testing to prevent catastrophic forgetting. However, these approaches often incur substantial computational overhead or fail to leverage prior knowledge effectively.

There are several settings in the field of CRL, which are mainly divided into three categories. First, the explicit task boundaries are required during both training and testing (Malagon et al., 2024). Second, the explicit task boundaries are not required during either training or testing (Rolnick et al., 2019; Oh et al., 2022). In this work, we focus on the third setting, which requires task boundary information during training but not during testing. This setting is widely adopted in CRL research (Pan et al., 2025; Powers et al., 2022a; Gaya et al., 2023; Schwarz et al., 2018).

## 3 METHOD

### 3.1 PRELIMINARIES

In general, reinforcement learning can be formulated as a sequence of Markov Decision Processes (MDPs). An MDP is defined as a framework in which an agent observes the current state $s$ of the environment, selects an action $a$, and receives a corresponding reward $r$ to the next state. Therefore, the MDPs can be formally represented as $\langle \mathcal{S}, \mathcal{A}, \mathcal{R}, \gamma \rangle$, where $\mathcal{S}$ denotes the state space, $\mathcal{A}$ the action space, $\mathcal{R}$ denotes the reward function, and $\gamma \in [0, 1]$ denotes the discount factor. Assuming

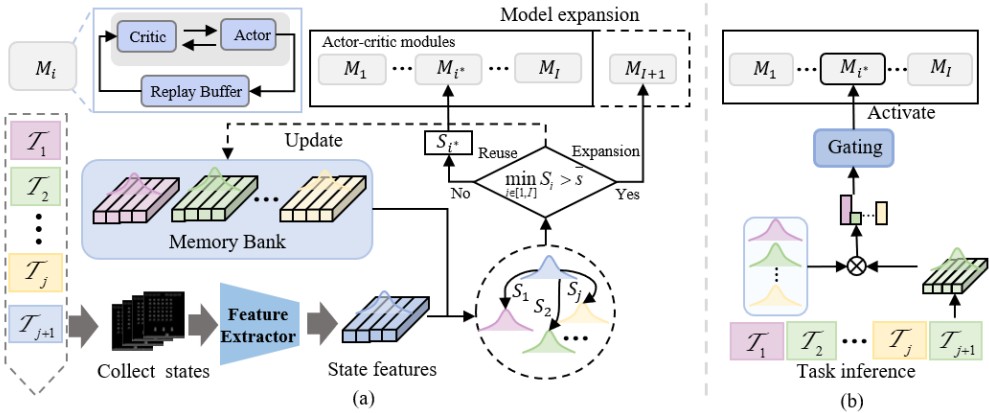

Figure 1: The framework of the proposed TADEN in dynamic environments. (a) Training pipeline; (b) Testing pipeline.

a total of $T$ time steps, the objective is to optimize the policy $\pi$ to maximize the cumulative reward $R = \sum_{t=1}^{T} \gamma r_t$ obtained over the entire process, where $r_t$ denotes the reward at the $t$-th step.

In CRL, non-stationary environments are typically modeled as sequences of MDPs, where both environmental dynamics and task characteristics change over time. We define a non-stationary task sequence as $\mathcal{T} = \{\mathcal{T}_1, \mathcal{T}_2, \cdots, \mathcal{T}_k, \cdots, \mathcal{T}_K\}$. Each task $\mathcal{T}_k$ is defined as a stationary MDPs $\langle \mathcal{S}^k, \mathcal{A}^k, \mathcal{R}^k, \gamma^k \rangle$. The agent is trained on each task for $T$ steps to maximize its cumulative reward by optimizing policy $\pi_k$ as follows:

$$\mathcal{L}_{\text{RL}} = \mathbb{E}_{\tau \sim \pi_k(\tau)} \left[ \gamma R(\tau) \right], \tag{1}$$

where the expectation is computed over the full trajectory $\tau$, which is generated by executing the policy $\pi_k$ from the initial state until the end of the agent's lifetime. During training, selecting an existing actor–critic module allows the resulting policy $\pi_k$ to be shared across multiple tasks, whereas selecting a new module produces a task-specific policy $\pi_k$.

## 3.2 TASK-AWARE EXPANSION STRATEGY

### 3.2.1 CONSTRUCTING TASK-SPECIFIC REPRESENTATION

In dynamic environments, the introduction of drastically new tasks often exacerbates the forgetting of previously learned tasks (Cai et al., 2021). Therefore, it is crucial to accurately identify the emergence of those tasks to initialize new policies and achieve effective parameter isolation. In this work, as shown in Fig. 1a, we collect a subset of environment observations before training each task to form a task-specific representation and store them in a memory bank, which guides subsequent module selection and policy adaptation.

To construct more informative task-specific representations, as illustrated in Fig. 1a, we first train an independent Proximal Policy Optimization (PPO) Algorithm RL agent in the environment for several steps before starting each task, to collect the observation sequence of the current task. To effectively leverage the knowledge contained in existing policies when optimizing new task policies, the similarity should be assessed using the observation sequences collected by a policy that is suitable for the new task rather than a random policy (Zhang et al., 2023). Therefore, our method calculates task similarity utilizing the collected environment observation sequence by the RL agent during training to determine whether the policy of existing tasks is suitable for new tasks. Therefore, our method computes task similarity using the environment observation sequences collected by the RL agent during training. This similarity assessment determines whether the policies from existing tasks apply to new tasks and whether to expand the network. Formally, for a given task $\mathcal{T}_k$, we first execute an RL method for $N$ steps to collect $N$ MDP tuples $\langle s^n, a^n, r^n, \gamma^n \rangle$, $n \in [1, N]$. The states collected $Q_k = \{s_k^1, s_k^2, \cdots, s_k^N\} \in \mathbb{R}^{N \times C \times H \times W}$ are processed through a feature extraction module to generate task-specific representations, denoted as $q_k \in \mathbb{R}^{N \times L}$, where $C$ represents the number of channel, $H$ and $W$ represent the size of the image, $L$ represents the length of the feature

vector. In RL, the state at the current time step is determined by the state and action of the previous time step. Therefore, the collected states $Q_k$ implicitly contain both visual and action information. The feature extraction step is formulated as:

$$q_k = f(Q_k), \tag{2}$$

where $f$ represents a pre-trained ResNet18 (He et al., 2016) on ImageNet as convolutional visual feature extractor. By having a set of $q_1, q_2, \ldots, q_k$, we can form a memory bank that encodes information for tasks. By storing the representation corresponding to old tasks in the memory bank, the similarity between a newly arrived task and the old tasks can be evaluated quantitatively.

### 3.2.2 TASK-AWARE EXPANSION

To alleviate performance degradation resulting from task conflicts, we employ dynamic expansion of the actor-critic modules, which enables parameter isolation for different tasks to reduce catastrophic forgetting. Specifically, when encountering a sequential task stream $\mathcal{T} = \{\mathcal{T}_1, \mathcal{T}_2, \cdots, \mathcal{T}_k, \cdots, \mathcal{T}_K\}$, we instantiate the initial actor-critic module $M_1$ to train the first task in the sequence. From the second task, we calculate the Wasserstein distance between the task-specific representation of the current task and the stored representations of previous tasks within the memory bank to assess task similarity. If the current task exhibits low similarity to all previously encountered tasks, a new actor-critic module is instantiated. Otherwise, the task is trained using the module associated with the most similar task, and its task-specific representation in the memory bank is accordingly updated. By expanding network modules, the method enables parameter isolation across tasks to alleviate catastrophic forgetting. Besides, reusing network modules enables weight sharing across tasks to facilitate effective knowledge transfer.

Formally, for $\mathcal{T}_k$, assuming that there are already $I$ actor-critic modules $\mathcal{M} = \{M_1, M_2, \cdots, M_I\}$ with their corresponding task-specific representations. We first normalize the task-specific representation $q_k$ and each existing task-specific representation $q_i \in \mathbb{R}^{N \times L}$ within the memory bank along the temporal dimension. Then split the representation into $L$ distributions to compute the Wasserstein distance (He et al., 2022) $\phi$ between $q_k$ and $q_i$ along that dimension. The overall similarity score $S_i$ is obtained by averaging the Wasserstein distances as follows:

$$S_i = \frac{1}{L} \sum_{l=1}^{L} \phi(\text{norm}[q_k(l)], \text{norm}[q_i(l)]), \forall i \in \{1, 2, \cdots, I\}, \tag{3}$$

where norm denotes the normalization operation (Ramdas et al., 2017), $q_k(l)$ denotes the $l$-th distribution of $q_k$.

Based on the task similarity score $S = \{S_1, S_2, \cdots, S_I\}$ which compares $\mathcal{T}_k$ with old tasks, task-aware expansion is performed by:

$$\begin{cases} \text{create } M_{I+1}, & \text{if } \min_{i \in [1,I]} S_i > \bar{s}, \\ \text{reuse } M_{i^*} \text{ with } i^* = \arg\min_{i \in [1,I]} S_i, & \text{otherwise}, \end{cases} \tag{4}$$

where $\bar{s}$ denotes the threshold, $M_{i^*}$ represents the actor-critic module corresponding to the most similar task. The parameter analysis is shown in Appendix E If an existing module is reused for the current task, its corresponding task-specific representation in the memory bank is updated accordingly as follows:

$$q_{i^*} = \lambda\, q_k + (1 - \lambda)\, q_{i^*}, \tag{5}$$

where $\lambda$ is a hyperparameter. In addition, to mitigate knowledge forgetting during the reuse of existing modules caused by new task training, we employ an experience replay mechanism (Rolnick et al., 2019). By dynamically expanding the network, task interference can be effectively mitigated, thereby preventing catastrophic forgetting caused by significant task conflicts.

### 3.3 DUAL-MODE INITIALIZATION

When we introduce a new actor-critic module during training, an appropriate initialization method is critical. Specifically, a naive random initialization offers better plasticity (Dohare et al., 2024; Abbas

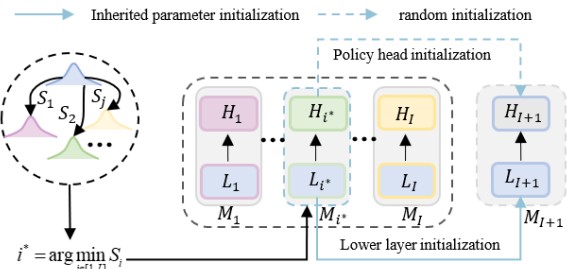

Figure 2: Extension module initialization.

et al., 2023), however, it cannot exploit prior knowledge. Conversely, initializing with the existing modules may lead to limited plasticity, impeding the learning of the new task. Therefore, we propose a dual-mode initialization technique that separates the actor-critic module $M_{i*}$ into top and lower layers $H_{i*}$ and $L_{i*}$, and initializes each independently. As shown in Fig. 2, when introducing a new actor-critic module $M_{I+1}$, we identify the most similar existing module $M_{i*}$ based on the similarity between representations and initialize the lower layers $L_{I+1}$ of the new module with the parameters of the selected $L_{i*}$, enabling knowledge transfer from previously learned tasks. Meanwhile, the top layer, denoted as the policy head $H_{I+1}$ is randomly initialized to allow flexible adaptation to the new task. This design allows the model to exploit existing knowledge and maintain plasticity for new tasks.

### 3.4 TASK INFERENCE

Real-world scenarios often lack explicit task labels and clear task boundaries, making it difficult for architecture-based methods to infer task identities at test time. This ambiguity hinders proper module selection and decision-making. In our work, we test all tasks, including both seen and unseen tasks, after training on each task. For each test task, the most appropriate network module is selected for testing. Specifically, when testing a task, we first run the agent using an RL policy for a few steps to collect observations and extract task-specific features. We then compute the Wasserstein distance between these representations and existing task representations in the memory bank as the Equation 3. Then the most appropriate network module is activated for testing as shown in Fig. 1b. The process is as follows:

$$M_{\text{test}} = \text{Activate}\,(M_{i*})\ \text{with}\ i^* = \arg\min_{i\in[1,I]} S_i. \tag{6}$$

where Activate denotes using the selected module for testing.

## 4 EXPERIMENTAL DESIGN

### 4.1 ENVIRONMENTS

We evaluated our proposed method in the MiniHack (Samvelyan et al., 2021) and Atari (Bellemare et al., 2012) environments, and compared its performance to several popular CRL methods.

#### 4.1.1 MINIHACK ENVIRONMENT.

MiniHack is built on the NetHack Learning Environment (Samvelyan et al., 2021), and offers a rich interaction interface for agent training. In this study, we focused on its navigation tasks as representative CRL challenges. MiniHack navigation tasks require the agent to reach a target location while overcoming diverse challenges, such as battling monsters in corridors, avoiding traps, and traversing complex mazes. To evaluate sequential learning capabilities, we selected 10 tasks from the Mini-Hack navigation suite and trained the agent on them sequentially. The agent receives rewards or penalties depending on its behavior, with the full reward granted only upon reaching the target location. A comprehensive description of the MiniHack tasks utilized in this study can be found in **Appendix** A.

#### 4.1.2 ATARI ENVIRONMENT.

The Atari game environment (Bellemare et al., 2012) is a widely used benchmark in RL, comprising a diverse set of classic arcade games, such as Pong, Breakout, and Space Invaders, each posing

unique challenges. In previous studies (Rolnick et al., 2019; Schwarz et al., 2018), six Atari games were used to evaluate CRL performance, including SpaceInvaders, Krull, BeamRider, Hero, Star-Gunner, and MsPacMan. Therefore, we also adopt these six Atari games to evaluate the performance of CRL methods. In Atari games, the agent selects actions, such as moving left or right, firing, and others, based on the observed environment state. At each time step, the agent receives a reward that reflects the outcome of its action within the current game context. A comprehensive description of the Atari tasks utilized in this study can be found in **Appendix** A.

## 4.2 EVALUATION METRICS

**Average Performance (AP):** We measure overall performance by calculating the average final reward obtained on all tasks. The detailed computation is as follows:

$$P = \frac{1}{n} \sum_{i=1}^{n} R_{i,\text{final}}, \tag{7}$$

where $n$ represents the total number of tasks to be trained, $R_{i,\text{final}}$ represents the reward obtained by evaluating the $i$-th task after training all tasks.

**Average Forgetting (AF):** This metric reflects the degree of knowledge forgetting on previously learned tasks after the agent is trained on subsequent tasks. According to recent studies (Wołczyk et al., 2021; Wang et al., 2024; Meng et al., 2025), the average forgetting metric is defined as follows:

$$F = \frac{1}{n-1} \sum_{i=1}^{n-1} (R_{i,i} - R_{i,\text{final}}), \tag{8}$$

where $R_{i,i}$ represents the reward of the $i$-th task obtained after training on the same task. With a similar AP value. A method having a lower AF value is better.

**Average Transfer (AT):** This metric assesses the extent to which knowledge from previously learned tasks facilitates the learning of new tasks. The average transfer metric is defined as follows:

$$T = \frac{1}{n-1} \sum_{i=2}^{n} (R_{i,i} - R_{i}^{\text{ind}}), \tag{9}$$

where $R_{i}^{ind}$ represents the reward of an independent model trained only on the $i$-th task. A method having a higher AT value is better.

Among the three metrics, AP is the most important one, as it directly reflects how well a method performs on all tasks at the end of CRL training. In fact, it includes the factors of forgetting and transferability which AF and AT aim to quantify. AF and AT measure the relative ability of a method across tasks and are only meaningful when AP is high. They serve as an auxiliary metric. It is possible that AF is quite low, but AP is also very low. E.g., a method performs poorly on all tasks, but it forgets little about the knowledge of any task. Such a method is useless. Similar arguments apply to AT.

## 4.3 METHODS FOR COMPARISON

FT: A single model is that fine-tuned sequentially across the entire task sequence during training. EWC (Kirkpatrick et al., 2017): This regularization-based method mitigates forgetting by constraining parameter updates, thereby preserving knowledge acquired from previous tasks throughout the training sequence. CLEAR (Rolnick et al., 2019): This replay-based method mitigates forgetting by storing data from previous tasks and interleaving it with new task data during training, enabling the model to retain prior knowledge. P&C (Schwarz et al., 2018): This method combines EWC-based regularization with policy distillation, transferring new task policies into a larger network to preserve knowledge from previous tasks. MoE (Li et al., 2025): the network consists of 4 experts and uses EWC to achieve continual learning. SANE (Powers et al., 2022a): This architecture-based method selectively adds or merges network modules by comparing value estimates, and incorporates experience replay to retain knowledge from previous tasks.

Among the aforementioned methods, FT and CLEAR do not require task boundary information during either training or testing, whereas the remaining methods require task boundary information

Table 1: The results across all task sequences and methods. Metrics are reported as means ± standard deviations computed over three independent runs, with the best results highlighted in bold. The Para. means the parameters of the model.

| Methods | Para. (M) | MiniHack | | | Atari | | |
|---------|-----------|----------|----------|----------|----------|----------|----------|
| | | AP↑ | AF↓ | AT↑ | AP↑ | AF↓ | AT↑ |
| FT | 1.7/1.7 | -0.19±0.11 | 0.19±0.03 | -0.77±0.38 | 647.66±23.81 | 3359.72±587.41 | -2715.11±1401.06 |
| EWC | 1.7/1.7 | 0.26±0.19 | 0.16±0.09 | -0.03±0.25 | 449.39±71.55 | 319.08±79.55 | -5241.07±96.88 |
| MoE | 2.2/2.2 | 0.27±0.09 | 0.27±0.06 | 0.13±0.11 | 701.30±140.09 | 158.16±211.77 | -5683.56±1603.86 |
| P&C | 7.0/7.0 | 0.40±0.01 | 0.02±0.02 | -0.03±0.01 | 849.90±306.89 | **-10.12**±250.86 | -5621.04±1562.17 |
| CLEAR | 1.7/1.7 | 0.55±0.10 | 0.14±0.06 | 0.29±0.09 | 2328.72±80.55 | 354.00±266.68 | -3558.76 ±1279.53 |
| SANE | 10.2/10.2 | 0.35±0.14 | 0.24±0.12 | 0.12±0.25 | 1937.22±953.42 | 5412.34±2305.79 | 1026.73±1539.49 |
| TADEN | 6.8/10.2 | **0.62**±0.06 | **0.01**±0.03 | **0.40**±0.28 | **12357.01**±1329.43 | 133.35±342.00 | **8171.96**±1513.47 |

during training but not during testing. The previously mentioned method (Ahn et al., 2025) is not publicly available, and the code of method (Gaya et al., 2023) cannot be implemented. making them infeasible to reproduce.

## 4.4 IMPLEMENTATION DETAILS

All models used in the experiments were implemented using the PyTorch framework. In both the MiniHack and Atari environments, training was conducted for two epochs, with all tasks trained sequentially within each epoch. In the MiniHack environment, each task was trained 1e6 steps and tested over 10 episodes every 1e5 steps. In the Atari environment, each task was trained 1e7 steps and tested over 10 episodes every 1e6 steps. The average reward across episodes is reported as the evaluation metric. All experiments were conducted on an RTX 3080 Ti GPU. A comprehensive description of the experiment setting can be found in **Appendix** B.

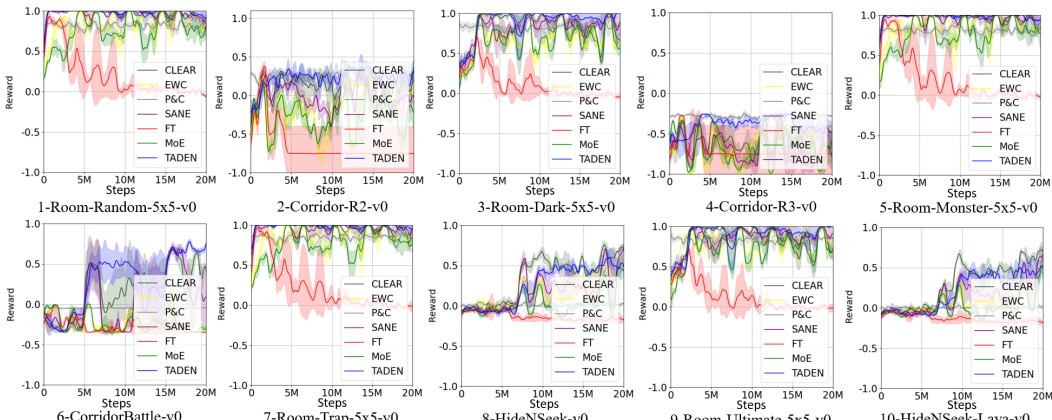

Figure 3: Testing curves of task average returns in the MiniHack environment. The first number in the task name under each panel indicates the task order during training. The training process consists of two epochs, each comprising a total of 10M steps. Every task is trained for 1M steps per epoch. All tasks were tested over 10 episodes every 0.1M steps during training, and the average reward across episodes was reported as the evaluation metric. The solid lines represent the mean average test returns, while shaded regions indicate the corresponding standard deviations, computed over three independent runs.

## 4.5 RESULTS

We report the performance on the MiniHack and Atari environments in Table 1. As explained in Evaluation Metrics before, we primarily focus on the AP metric. Only when two methods achieve similar AP values, we compare the auxiliary metrics AF and AT. From Table 1, it is seen that our proposed TADEN achieved the highest AP in the MiniHack environment, significantly surpassing all other methods. The replay-based method CLEAR ranked the second, with a performance lower by 0.07. In the Atari environment, TADEN attained the highest AP and consistently outperformed all baseline methods by a substantial margin. Moreover, compared to architecture-based methods such as P&C and SANE, TADEN achieved the highest AP with fewer parameters.

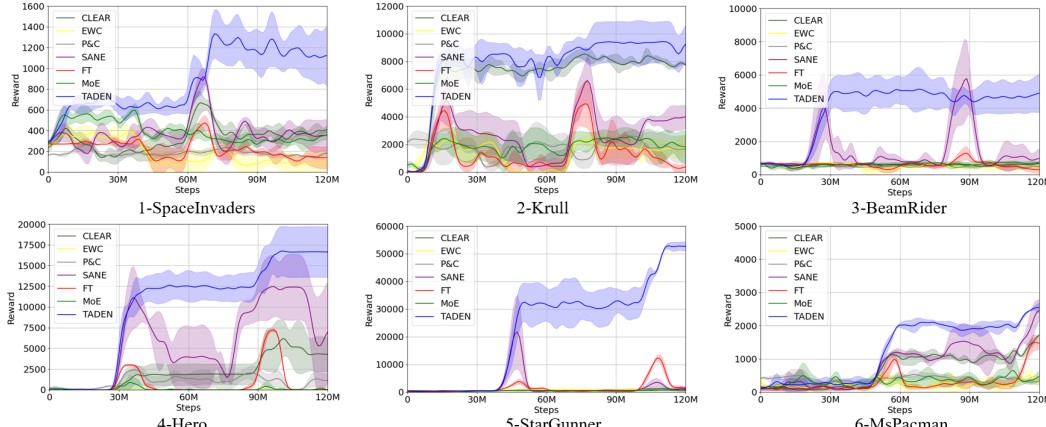

Figure 4: Testing curves of task average returns in the Atari environment. The first number in the task name under each panel indicates the task order during training. The training process consists of two epochs, each comprising a total of 60M steps. Every task is trained for 10M steps per epoch. All tasks were tested over 10 episodes every 1M steps during training, and the average reward across episodes was reported as the evaluation metric.

In terms of forgetting in the MiniHack environment, our proposed TADEN training framework also achieved the best AF value. In the Atari environment, since the AP of our TADEN was higher by a large margin than baseline methods, there was no need to compare the auxiliary metric AF. In fact, as seen in Table 1, P&C achieved a much lower AF value than other methods, but from Fig. 4, P&C yielded very low rewards on most tasks, rendering it largely ineffective.

In addition, Figs. 3 and 4 present the average episodic returns across all tasks in the MiniHack and Atari environments during the testing phase. As shown in Fig. 3, the proposed TADEN training framework consistently achieved better performance across the majority of tasks and effectively mitigated catastrophic forgetting in the MiniHack environment. On individual tasks such as tasks 8 and 10, although TADEN's performance was slightly lower than that of CLEAR and SANE, it demonstrated better stability. This indicated that the proposed method effectively alleviates knowledge forgetting through the dynamic expansion of network modules. In addition, as shown in Fig. 4, TADEN achieved better performance across all tasks. The final performance of each task is shown in the **Appendix** C.

After training, TADEN ultimately comprised four modules across the 10 MiniHack tasks and six modules across the six Atari tasks. As shown in Table 1, our TADEN achieved higher AP in the MiniHack environment compared to PC and SANE with fewer parameters. This demonstrates that our approach effectively mitigates the parameter growth typically associated with network expansion by selectively reusing existing modules across similar tasks. In the Atari environment, our method achieved a substantial improvement in AP while using the same number of parameters as SANE. This result demonstrates that the proposed dual-mode initialization effectively leverages existing knowledge to facilitate learning of new tasks. The module expansion process and time costs during training are shown in the **Appendix** D.

## 4.6 Ablation Study

### 4.6.1 Analysis of the Task-Aware Expansion

We adopt an RL approach to collect state features and obtain task-specific representations, which are used to compute the inter-task similarities by Wasserstein distance for guiding module expansion. To evaluate the effectiveness of this strategy, we compared it with two baseline methods: (1) using random sampling to collect state features for distribution estimation, and (2) computing the cosine similarity between the centroid vectors of state features. As shown in Fig. 5a,b, the proposed method achieved significantly better AP compared to the other two approaches. This demonstrates that merely sampling environment states at random to obtain task-specific representations cannot accurately determine whether the existing policy applies to new tasks. Besides, utilizing the centroid vector cannot obtain a good task representation and is not enough to accurately evaluate task similarity. These results prove that our method can effectively expand the network as task representations to calculate task similarity.

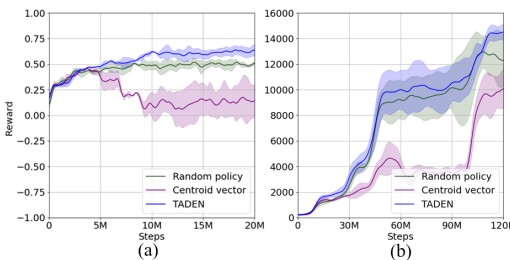 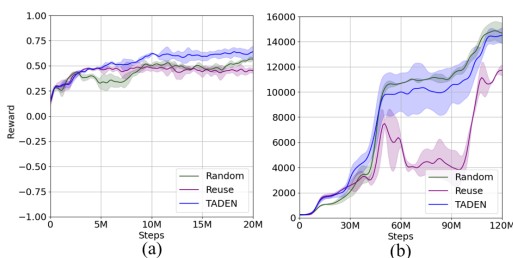

Figure 5: Average reward curves across all tasks with different module expansion strategies in the (a) MiniHack and (b) Atari environments. "Random Policy" denotes collecting observations randomly. "Centroid Vector" denotes computing the cosine similarity between the centroid vectors of environment state features.

Figure 6: Average reward curves across all tasks with different module initialization strategies in the (a) MiniHack and (b) Atari environments. "Random" denotes random initialization; "Reuse" refers to reusing existing modules for initialization.

### 4.6.2 Influence of the Initialization of New Module

To evaluate the effectiveness of our proposed dual-mode initialization strategy, we compared it with two approaches: random initialization of the entire module, initialization using the most similar existing module. The test average reward during training is shown in Fig. 6a,b. Our initialization method outperformed the other two approaches, achieving the best average performance in the MiniHack environment. Although direct random initialization provides better plasticity, it failed to utilize prior knowledge, which significantly hindered the acquisition of task-specific skills, especially in the case of complex or high-difficulty tasks. While initializing with the most similar existing module enables effective transfer of prior knowledge, the convergence of that module restricts the model's flexibility, thereby limiting its ability to adapt to novel task-specific features. To balance knowledge reuse and adaptability, we initialize the low layers of the new module using the most similar existing module, and randomly initialize its top layer. This strategy promotes both knowledge transfer and plasticity, resulting in enhanced overall model performance. In the Atari environment, random initialization slightly outperforms our proposed method. This may be attributed to the low correlation between Atari tasks, where initializing the underlying network with existing modules introduces little interference that can marginally degrade performance. Nevertheless, the overall average performance remains better than other baselines.

## 5 Limitation

The limitations of the proposed DATEN are as follows. First, TADEN requires a small amount of data for experience replay, which may pose risks of privacy leakage in sensitive applications. Moreover, although the network is dynamically expanded during training, when the number of tasks becomes large and inter-task conflicts are substantial, the continual growth of the network can lead to increased computational and memory overhead. In the future, we plan to investigate generative experience replay as a means of enhancing privacy protection. Additionally, we aim to incorporate regularization techniques to constrain the expansion of network modules, thereby further reducing computational overhead and improving scalability.

## 6 Conclusion

In this work, we propose the TADEN training framework, a CRL method that leverages task-aware dynamic network expansion to mitigate catastrophic forgetting in non-stationary environments. First, we utilize an RL method to collect state sequences. Then compute task similarities to dynamically determine whether to expand the network or reuse existing modules. During module expansion, we initialize the low layers of the new module with the most similar existing module and randomly initialize its top layer. This allows for leveraging prior knowledge while preserving the plasticity required for new task adaptation. Finally, we evaluated our proposed TADEN and demonstrated better average performance in both the MiniHack and Atari environments.

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

# A  ENVIRONMENTS

The experiment involves two task sequences from MiniHack and Atari, where agents are trained sequentially to achieve continual reinforcement learning.

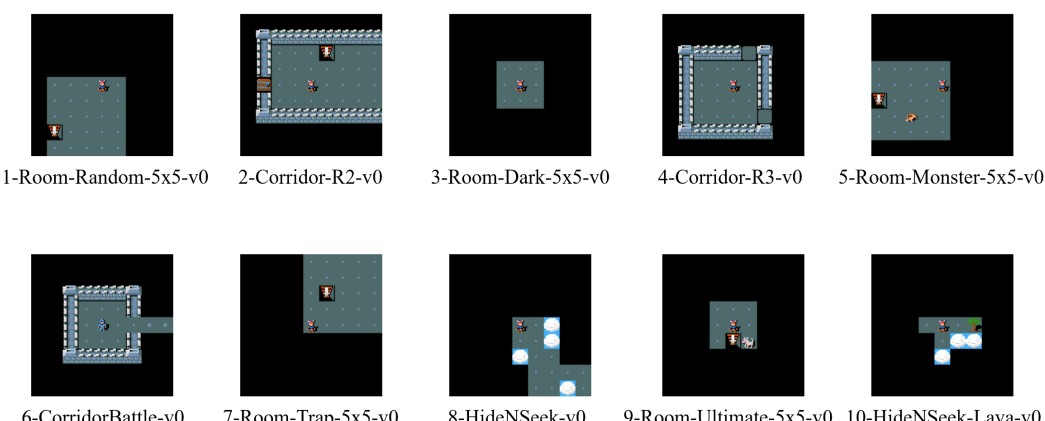

1-Room-Random-5x5-v0  2-Corridor-R2-v0  3-Room-Dark-5x5-v0  4-Corridor-R3-v0  5-Room-Monster-5x5-v0

6-CorridorBattle-v0  7-Room-Trap-5x5-v0  8-HideNSeek-v0  9-Room-Ultimate-5x5-v0  10-HideNSeek-Lava-v0

Figure S1: Examples of initial observations for each task in the MiniHack environment.

### A.0.1  MINIHACK ENVIRONMENT

We selected 10 tasks from the MiniHack environment to conduct continual learning experiments. The task sequence includes: (1) Room-Random-5x5-v0, (2) Corridor-R2-v0, (3) Room-Dark-5x5-v0, (4) Corridor-R3-v0, (5) Room-Monster-5x5-v0, (6) CorridorBattle-v0, (7) Room-Trap-5x5-v0, (8) HideNSeek-v0, (9) Room-Ultimate-5x5-v0, and (10) HideNSeek-Lava-v0. Fig. S1 presents the randomly initialized observations for each task, and we provide detailed descriptions of the task sequence below.

**1-Room-Random-5x5-v0:** The agent is required to explore a randomly generated room to reach the goal. In each episode, the layout, as well as the initial positions of the agent and the goal, are randomly initialized.

**2-Corridor-R2-v0:** The agent is required to reach the exit by navigating through two connected corridors, with the positions of the agent and the exit randomized in each episode.

**3-Room-Dark-5x5-v0:** The agent is required to find the goal hidden in a dark room, with both the agent's starting position and the goal location randomized in each episode.

**4-Corridor-R3-v0:** The agent is required to reach the exit by navigating through three connected corridors, with randomized agent and exit positions in each episode.

**5-Room-Monster-5x5-v0:** The agent is required to reach the goal while avoiding or defeating a monster in the room. The positions of the agent, monster, and goal are randomized in each episode.

**6-CorridorBattle-v0:** The agent is required to fight monsters in the corridor and navigate through it to reach the exit. The positions of the agent, enemies, and exit are randomized in each episode.

**7-Room-Trap-5x5-v0:** The agent is required to reach the goal while avoiding hidden traps scattered in the room. The positions of the agent and the goal are randomized in each episode.

**8-HideNSeek-v0:** The agent is required to find and reach the hidden target while avoiding detection. The positions of the agent and the goal are randomized in each episode.

**9-Room-Ultimate-5x5-v0:** The agent is required to reach the goal while navigating through a room filled with monsters and traps. The positions of the agent, monsters, traps, and the goal are randomized in each episode.

**10-HideNSeek-Lava-v0:** The agent is required to find and reach the hidden target while avoiding dangerous lava hazards. The positions of the agent, target, and lava are randomized in each episode.

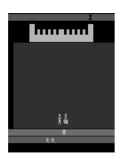 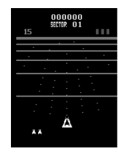 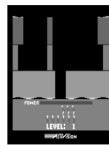 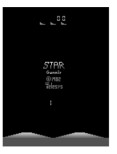 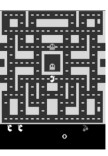

1-SpaceInvaders  2-Krull  3-BeamRider  4-Hero  5-StarGunner  6-MsPacman

Figure S2: Examples of initial observations for each task in the Atari environment.

### A.0.2 ATARI ENVIRONMENT

We selected 6 tasks from the Atari environment to conduct continual learning experiments. The task sequence includes: (1) SpaceInvaders, (2) Krull, (3) BeamRider, (4) Hero, (5) StarGunner, and (6) MsPacMan. Fig. S2 presents the randomly initialized observations for each task, and we provide detailed descriptions of the task sequence below.

**1-SpaceInvaders:** The agent is required to move horizontally to shoot descending aliens. The goal is to eliminate as many aliens as possible while avoiding enemy fire. The positions of the aliens and the agent are randomized in each episode.

**2-Krull:** The agent is required to navigate a landscape to defeat enemies and rescue a captive. The positions of enemies and obstacles are randomized each episode. The agent must avoid hazards while attacking foes to progress.

**3-BeamRider:** The agent is required to shoot down waves of enemy ships while avoiding their attacks. Enemy positions and attack patterns are randomized each episode. The goal is to survive and maximize the score.

**4-Hero:** The agent is required to navigate through a castle to rescue a princess. The environment contains enemies and traps with randomized positions in each episode. The agent must avoid dangers and defeat foes to reach the goal.

**5-StarGunner:** The agent is required to shoot down enemy ships while avoiding incoming attacks. Enemy spawn locations and attack patterns are randomized each episode. The goal is to survive and eliminate as many enemies as possible.

**6-MsPacMan:** The agent is required to navigate a maze to eat all pellets while avoiding ghosts. The positions and movements of ghosts are randomized each episode. The goal is to clear the maze without being caught.

## B  DETAILS ON EXPERIMENTS

### B.1  NETWORK ACHITECTURE

In our framework, the actor-critic module $M_i$ consisted of two separate neural networks: an actor network and a critic network. As illustrated in Fig. S3, the actor network consisted of three convolutional (CNN) layers followed by two fully connected (FC) layers. The final linear layer of the actor network outputted a probability distribution over the action space. We utilized the MiniHack and Atari game image with the dimension of $1 \times 3 \times 84 \times 84$ as input of the lower layer $L_i$, after the first CNN layer with kernel size 8 and stride 4, the second CNN layer with kernel size 4 and stride 2, the last CNN layer with kernel size 3 and stride 1, and the first FC layer to obtain the feature vector with the dimension of 512. Then, the feature vector was input to the top layer (policy head), which consists of an FC layer, and obtained the probability distribution over the action space. The critic network shared the same architectural design as the actor network, except for the final linear layer, which produced a scalar value representing the estimated value. In our work, when expanding the network, the CNN layers and the first FC layer of the actor-critic module were used as the lower layer and initialized using the existing modules, while the last FC layer was used as the policy head and randomly initialized.

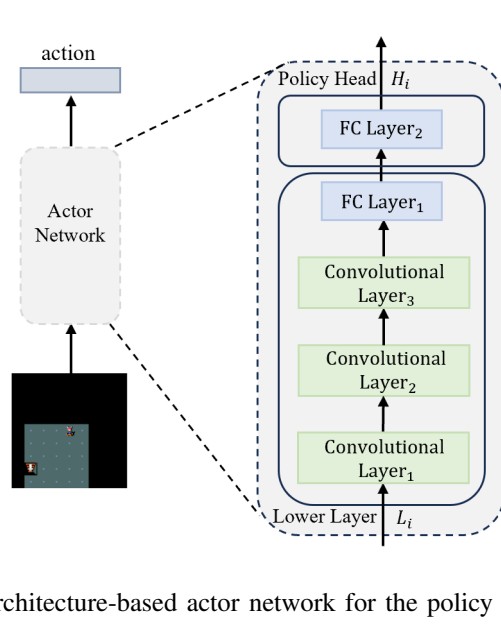

Figure S3: The CNN architecture-based actor network for the policy of the MiniHack and Atari tasks.

## B.2 HYPERPARAMETERS

In our experiments, all reinforcement learning agents were trained using an IMPALA-based training framework in both the MiniHack and Atari environments for 2 epochs. In MiniHack, each task was trained for 1e6 steps per epoch, with evaluations conducted every 1e5 steps to monitor reward performance. In the Atari environment, each task was trained for 1e7 steps per epoch, and evaluations were performed every 1e6 steps. The training hyperparameters for our proposed method in the MiniHack and Atari environments were summarized in Table S1.

Table S1: The hyperparameters of our proposed method in the task sequences of MiniHack and Atari environments.

| Hyperparameters | Ours |
| --- | --- |
| Num. actors | 64 |
| Learner threads | 2 |
| Batch size | 32 |
| Unroll length | 25 |
| Grad clip | 40 |
| Entropy cost | 0.001 |
| Discount factor | 0.99 |
| Learning rate | 4.8e-6 |
| Replay buffer size | 2e5 |
| Policy cloning weight | 0.01 |
| Value cloning weight | 0.005 |
| Similarity threshold | 0.28 |

## C THE FINAL REWARD OF MINIHACK AND ATARI TASKS

The final rewards obtained on each task by all baseline methods and our proposed TADEN training framework in MiniHack and Atari environments are reported in Tables S2 and S3. Table S2 shows that the proposed method TADEN achieved competitive performance, with an average performance of 0.63, exceeding the second-best by 0.09. In addition, on 10 tasks, our method achieved the best performance in 1-Room-Random-5x5-v0, 3-Room-Dark-5x5-v0, 6-CorridorBattle-v0, and 9-Room-Ultimate-5x5-v0, and the second best in 2-Corridor-R2-v0, 5-Room-Monster-5x5-v0, and 7-Room-Trap-5x5-v0. Moreover, our proposed TADEN training framework obtained the best final

reward on all Atari tasks as shown in Table S3, indicating that our task-aware expansion strategy and dual-mode initialization method effectively expand the module and alleviate the catastrophic forgetting.

Table S2: The final rewards of different individual tasks in the MiniHack environment.

| Tasks | EWC | MoE | P&C | CLEAR | SANE | FT | Ours |
|---|---|---|---|---|---|---|---|
| 1 | 0.90±0.00 | 0.83±0.21 | 0.88±0.00 | 0.82±0.06 | 0.82±0.25 | -0.10±0.00 | 0.96±0.06 |
| 2 | 0.23±0.11 | -0.19±0.40 | 0.22±0.16 | 0.51±0.27 | 0.01±0.55 | -0.75±0.43 | 0.26±0.11 |
| 3 | 0.05±0.31 | 0.62±0.05 | 0.85±0.15 | 0.63±0.17 | 0.89±0.19 | -0.03±0.13 | 1.00±0.00 |
| 4 | -0.47±0.04 | -0.65±0.23 | -0.29±0.04 | -0.60±0.07 | -0.91±0.08 | -0.75±0.43 | -0.54±0.17 |
| 5 | 0.93±0.060 | 0.76±0.26 | 0.81±0.12 | 0.89±0.11 | 0.96±0.06 | -0.03±0.06 | 0.93±0.12 |
| 6 | -0.32±0.03 | -0.31±0.05 | -0.040±0.01 | 0.36±0.27 | 0.01±0.49 | -0.35±0.00 | 0.87±0.15 |
| 7 | 0.86±0.06 | 0.83±0.21 | 0.78±0.00 | 0.78±0.22 | 1.0±0.00 | 0.01±0.11 | 0.93±0.13 |
| 8 | 0.62±0.06 | 0.43±0.05 | -0.01±0.00 | 0.70±0.20 | 0.73±0.06 | -0.17±0.06 | 0.43±0.06 |
| 9 | 0.69±0.10 | 0.75±0.26 | 0.81±0.06 | 0.64±0.07 | 0.85±0.13 | -0.03±0.06 | 1.0±0.00 |
| 10 | 0.60±0.10 | 0.42±0.33 | 0.03±0.06 | 0.66±0.15 | 0.73±0.12 | -0.18±0.04 | 0.46±0.31 |
| Average | 0.45±0.04 | 0.35±0.12 | 0.40±0.02 | 0.54±0.08 | 0.50±0.07 | -0.24±0.07 | **0.63**±0.09 |

Table S3: The final rewards of different individual tasks in the Atari environment.

| Tasks | EWC | MoE | P&C | CLEAR | SANE | FT | Ours |
|---|---|---|---|---|---|---|---|
| 1 | 77.00 ±110.64 | 377.83 ±131.63 | 193.83 ±45.87 | 412.67 ±152.34 | 359.00 ±185.33 | 160.00 ±140.00 | 1073.17 ±459.33 |
| 2 | 1867.67 ±759.75 | 1724.67 ±1281.03 | 1691.00 ±1214.49 | 7431.67 ±421.29 | 3956.33 ±947.06 | 409.67 ±384.07 | 9632.00 ±1836.05 |
| 3 | 384.93 ±266.74 | 750.40 ±88.06 | 452.67 ±38.33 | 685.60 ±133.22 | 1036.07 ±885.45 | 309.47 ±299.24 | 5013.27 ±1549.63 |
| 4 | 0 | 0 | 1249.33 ±191.76 | 4278.33 ±2574.07 | 7534.33 ±7400.35 | 0 | 16660.16 ±3744.36 |
| 5 | 940.00 ±124.90 | 666.67 ±387.34 | 850.00 ±186.81 | 1143.33 ±652.10 | 640.00 ±271.85 | 1326.67 ±549.30 | 52703.33 ±2025.35 |
| 6 | 316.67 ±130.24 | 564.00 ±162.89 | 329.33 ±194.63 | 1838.00 ±164.76 | 2394.33 ±390.49 | 1330.67 ±202.24 | 2535.33 ±232.42 |
| Average | 597.71 ±122.30 | 680.60 ±158.04 | 794.19 ±408.31 | 2631.60 ±429.89 | 2653.34 ±1182.03 | 589.41 ±62.85 | **14557.878** ±895.17 |

## D    NETWORK EXPANSION DURING TRAINING

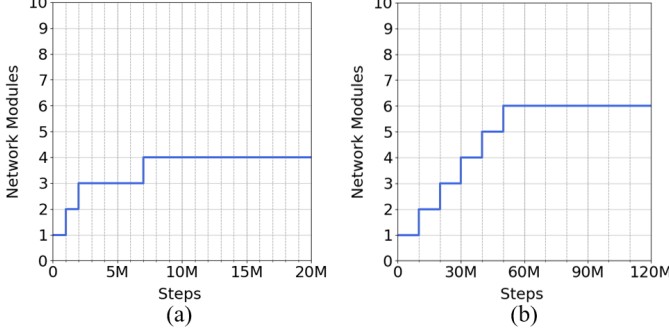

Figure S4: The network expansion during training in the (a) MiniHack and (b) Atari environments. The grid between the dotted lines represents the single-task training process.

The network expansion of our proposed TADEN during the training process in the MiniHack and Atari environments is shown in Fig. S5. In Fig. S5a, after training on 10 MiniHack tasks, TADEN

included four actor-critic modules. During training, task similarity guided the selection of either reusing existing modules or expanding new ones. Reusing modules for similar tasks reduced network size and resource consumption, while extending the network when there is task conflict can mitigate the catastrophic forgetting. For example, in Fig. S5a, a network module $M_2$ was added when training on task 2-Corridor-R2-v0. For tasks 4-Corridor-R3-v0 and 6-CorridorBattle-v0, the module $M_2$ was reused instead of expanding the network, as these tasks exhibit high similarity with task 2-Corridor-R2-v0. In Fig. S5b, after training on 6 Atari tasks, TADEN included six actor-critic modules, likely due to the low task correlation in the Atari environment. The results demonstrate that the proposed TADEN can dynamically expand network modules based on task similarity. Existing modules were effectively reused when encountering similar tasks to limit the growth of the overall network size.

During the training process, the training time for a single task is 1.0 hours in the MiniHack environment and 1.5 hours in the Atari environment.

## E  PARAMETERS ANALYSIS

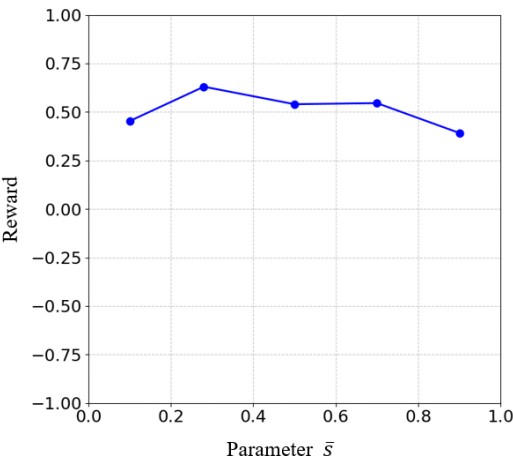

Figure S5: The parameter $\bar{s}$ analysis in MiniHack environment.

The parameter $\bar{s}$ served as the task similarity threshold to control the degree of network expansion during training. As shown in the figure, the model achieved its optimal performance at 0.28. This result demonstrated that our method reused existing modules for similar tasks while allocating new modules for more diverse tasks to mitigate catastrophic forgetting.

The parameter $\lambda$ served as the weight to fuse the features of similar tasks. Because the parameter shows little sensitivity to performance, it is set to 0.5 throughout our experiments.

## F  THE USE OF LARGE LANGUAGE MODELS (LLMS)

In the preparation of this paper, we utilized ChatGPT for language polishing and refinement. The model was used solely to improve the clarity, coherence, and fluency of the text. We remain solely responsible for the content, ideas, and integrity of the work.