# OpenReview forum: "A Task-Aware Dynamic Expansion Network for Continual Reinforcement Learning"
_ICLR.cc/2026/Conference — ICLR 2026 Conference Withdrawn Submission_

### Official Review · Reviewer_n2uM · 2025-10-18

**Soundness:** 3
**Presentation:** 3
**Contribution:** 2
**Rating:** 4
**Confidence:** 3

**Summary:**

This paper introduces the Task-Aware Dynamic Expansion Network (TADEN), a novel framework designed to address a critical challenge in Continual Reinforcement Learning (CRL): catastrophic forgetting. The TADEN framework is built on several key components:

**Task-Specific Representation and Similarity Measurement:** The core idea is to define a task's "identity" by the statistical distribution of environment states an agent encounters while performing that task. Before fully training on a new task, TADEN uses a reinforcement learning policy (PPO) for a few steps to collect a sequence of states. This sequence is passed through a feature extractor (a pre-trained ResNet18) to create a feature distribution that represents the task. This representation is then compared to the representations of previously learned tasks (stored in a memory bank) using the Wasserstein distance, a metric robust for comparing distributions that may not overlap.

**Task-Aware Expansion and Reuse:**The calculated task similarity score serves as the basis for an architectural decision. If the similarity score for a new task is low compared to all previous tasks (i.e., the minimum Wasserstein distance is above a set threshold), the new task is deemed dissimilar. In this case, TADEN expands the network by creating a new, separate actor-critic module to prevent interference and protect existing knowledge. Conversely, if the task is found to be similar to a previous one, the framework reuses the existing module associated with the most similar task, promoting knowledge sharing and limiting model growth.

**Dual-Mode Initialization:** When a new module is created, TADEN employs a hybrid initialization strategy to balance knowledge transfer and adaptability. The lower layers of the new module, which are responsible for feature extraction, inherit their parameters from the most similar existing module. This facilitates the transfer of general, low-level knowledge. The top layer, or "policy head," which is responsible for final decision-making, is randomly initialized. This ensures the module has the plasticity required to adapt to the unique demands of the new task.

**Task Inference at Test Time:** A crucial feature of TADEN is its ability to operate without explicit task labels during testing. When faced with a test environment, the agent collects a small number of observations, computes the task representation, and compares it to the representations in its memory bank. The module corresponding to the most similar task is then activated to perform the task, enabling the agent to autonomously select the correct skill for the job.

**Strengths:**

1. Clear Writing

2. Clear Motivation

3. Outperforming Performance

**Weaknesses:**

1. Insufficient Reference Check: There are some context-aware methods in reinforcement learning, e.g., [1]. Although they did not focus on continual RL, it is necessary to explain previous context-aware methods.

2. Lack of comparison with state-of-the-art methods: For instance, although the authors mentioned that the public code in [1] is not available, but I found that they released a public code.

---


[1]  Lee, Kimin, et al. "Context-aware dynamics model for generalization in model-based reinforcement learning." International Conference on Machine Learning. PMLR, 2020.

[2]  Ahn, Hongjoon, et al. "Prevalence of negative transfer in continual reinforcement learning: Analyses and a simple baseline." The Thirteenth International Conference on Learning Representations. 2025.

**Questions:**

1. It seems that similar task-aware dynamic approaches have already been developed, e.g., [1]. I suggest that the authors include a discussion of these prior methods and clarify what specifically distinguishes their proposed approach.

2. The baselines appear to be rather classical. The MoE method is not tailored for continual reinforcement learning but is instead a standard continual learning approach. Moreover, BC-based methods are not included in the comparison. Additionally, although the authors mentioned that the public code in [2] is unavailable, I found that it has actually been released. I recommend that the authors explicitly clarify this point and demonstrate that their proposed method remains effective even when compared to other relevant approaches not covered here.

3. The authors state that previous network expansion methods incur high memory and computational costs. Could the authors elaborate on how their method addresses these issues in terms of network expansion efficiency? Although Section D provides a partial explanation, it would be very helpful if the authors could also discuss the general tendencies of other architecture-based methods for comparison.


[1]  Lee, Kimin, et al. "Context-aware dynamics model for generalization in model-based reinforcement learning." International Conference on Machine Learning. PMLR, 2020.

[2]  Ahn, Hongjoon, et al. "Prevalence of negative transfer in continual reinforcement learning: Analyses and a simple baseline." The Thirteenth International Conference on Learning Representations. 2025.

---

### Official Review · Reviewer_mDhk · 2025-10-18

**Soundness:** 1
**Presentation:** 2
**Contribution:** 1
**Rating:** 2
**Confidence:** 5

**Summary:**

This paper introduces TADEN (Task-Aware Dynamic Expansion Network) for continual reinforcement learning. The method keeps a memory bank of task representations built from short sequences of observations collected by briefly running a policy in each new environment and encoding them with a fixed ResNet-18. For a new task, it compares the representation to stored ones using the Wasserstein distance to decide whether to reuse an existing module or create a new one. New modules reuse lower layers from the closest module while randomly initializing the policy head for adaptation. At test time, the model identifies the right module by comparing a few new observations with the memory bank. Experiments on MiniHack and Atari show that TADEN outperforms previous continual reinforcement learning baselines, with ablations confirming the value of its expansion rule and initialization design.

**Strengths:**

- Easy to follow and read.
- Empirical results on the Atari benchmark show that TADEN effectively prevents forgetting during continual learning.

**Weaknesses:**

- The framework combines known ideas, such as Meta-RL style task embedding construction [1,2,3] for the memory bank and model expansion [4,5], but it does not introduce a fundamentally new learning principle.
- Ambiguity in the paper’s claim and problem formulation. For example, the abstract states that “...struggle to preserve previously learned skills...,” but the actual method operates based on task-level actor critic moduleestimation. There is a lack of concrete evidence of how skills are preserved in practice .The paper should report an analysis of the correlation between task similarity and correct actor critic moduleretrieval, ideally by comparing the retrieved model context with the true task context. as the knowledge sharing occurs across experts or actual skills.
- Lack of fairness in comparison. One of the easiest ways to address the stability–plasticity trade-off is to use more memory (e.g., replay buffer or model parameters). At minimum, evaluations should be conducted under the same memory budget. In the current context, TADEN uses both but does not justify this choice. To support its claims, there must be evidence that TADEN is efficient even under this setup, and such evidence should be organically connected to the main argument in the introduction. The ablation in Appendix E(parameters analysis) alone is insufficient to evaluate this point.
- Limited generalization and restrictive assumptions in experiments. The reported improvements primarily appear in Atari tasks, where task similarity is visually distinguishable. However, Appendix D, Figure S4(b), shows that new tasks are almost perfectly isolated during the first epoch (first loop of six tasks). For instance, SpaceInvaders-1 exhibits performance gains only between 0M–10M and 60M–70M steps during the first epoch, while remaining stable otherwise.
- The paper should analyze how well task similarity aligns with correct actor critic moduleretrieval, ideally by examining whether the retrieved model truly matches the underlying task context. It should also clarify whether the performance mainly depends on task retrieval accuracy or actual knowledge sharing.
- Figure 4 shows that performance improvement occurs only at the points where each task is being trained during the repeated continual scenario (epochs). The authors should transparently report how close this performance is to the oracle architecture that uses true task IDs.

[1] Rakelly, Kate, et al. "Efficient off-policy meta-reinforcement learning via probabilistic context variables." International conference on machine learning. PMLR, 2019.

[2] Zintgraf, Luisa, et al. "Varibad: Variational bayes-adaptive deep rl via meta-learning." Journal of Machine Learning Research 22.289 (2021): 1-39.

[3] Duan, Yan, et al. "Rl $^ 2$: Fast reinforcement learning via slow reinforcement learning." arXiv preprint arXiv:1611.02779 (2016).

[4]  Aljundi, Rahaf, Punarjay Chakravarty, and Tinne Tuytelaars. “Expert Gate: Lifelong Learning with a Network of Experts.” *Proceedings of the IEEE Conference on Computer Vision and Pattern Recognition (CVPR)*, 2017

[5] Yoon, Jaehong, Eunho Yang, Jeongtae Lee, and Sung Ju Hwang. *“Lifelong Learning with Dynamically Expandable Networks.”* *International Conference on Learning Representations (ICLR)*, 2018.

**Questions:**

- It appears that the evaluation uses an action head customized for each environment. In this case, the evaluator effectively knows which environment is being tested, which also indicates the corresponding task head. Could the authors clarify why the task ID is not explicitly used during evaluation?
- Do common skills genuinely transfer across tasks? It would be helpful if the authors could provide supporting evidence, perhaps from the MiniHack experiments, demonstrating that meaningful knowledge sharing takes place.
- TADEN uniquely collects additional RL trajectories solely for constructing its task embeddings(memory bank). Is it fair to compare its performance with other methods that do not perform such extra data collection?

---

### Official Review · Reviewer_z4qT · 2025-10-20

**Soundness:** 2
**Presentation:** 2
**Contribution:** 2
**Rating:** 2
**Confidence:** 4

**Summary:**

This paper introduces the Task-Aware Dynamic Expansion Network (TADEN), a continual learning framework that first estimates inter-task similarity from collected state representations and then uses this estimate to decide whether to expand model capacity. When expansion is warranted, TADEN leverages prior knowledge while preserving adaptability by initializing new modules with reused lower-layer parameters from existing modules. Empirical evaluations on the MiniHack and Atari benchmarks demonstrate the effectiveness of the approach.

**Strengths:**

1.	The manuscript is clearly written and well structured.
2.	The proposed method is easy to understand.
3.	The experimental results are promising.

**Weaknesses:**

1.	The current set of baselines appears outdated relative to the paper’s emphasis on architectural expansion. Given that TADEN is an expansion-based approach, the evaluation should foreground contemporaneous (2024–2025) capacity-growth and modular-expansion methods. Relying primarily on SANE (2022) and a 2025 MoE variant that combines EWC with MoE may not sufficiently represent the current state of the art.
2.	On Atari, random initialization outperforms the proposed knowledge-transfer initialization, which calls into question the claimed benefits of transfer—the paper’s central motivation. What’s more, a more complete ablation would compare: (i) random initialization; (ii) initialization from the most similar existing module; (iii) initialization from a randomly chosen prior task/module.
3.	The empirical difference between Reuse and TADEN requires deeper analysis, as their architectures differ only in the initialization of the top layer. If, as the authors suggest, the inferior performance of Reuse stems from “convergence restricting the model’s flexibility,” this could potentially be mitigated by tuning the learning rate schedule or employing adaptive optimizers. A systematic study of these factors would strengthen the causal interpretation.
4.	The manuscript does not clearly articulate how previously acquired knowledge is maintained or utilized during subsequent training phases. It remains unclear whether any explicit regularization mechanisms are employed to prevent catastrophic forgetting, such as constraining the update magnitude of low-level feature representations.
5.	The approach’s behavior when facing dissimilar or non-transferable tasks is insufficiently analyzed. In such scenarios, does the framework degenerate into training an independent network per task, with the memory bank serving merely as a task identifier?
6.	While the paper focuses on continual RL, the proposed components—namely the memory bank and model expansion—could also apply to general continual learning settings. It remains unclear whether any module is specifically tailored for the RL context.

**Questions:**

See the weakness.

---

### Official Review · Reviewer_KJZp · 2025-10-31

**Soundness:** 3
**Presentation:** 4
**Contribution:** 3
**Rating:** 8
**Confidence:** 2

**Summary:**

The paper introduces TADEN, a **Task-Aware Dynamic Expansion Network** designed to alleviate catastrophic forgetting in Continual Reinforcement Learning (CRL). The key innovation is a task similarity-driven network expansion mechanism: the agent measures similarity between tasks using *Wasserstein distance* on environment state embeddings and decides whether to reuse existing actor–critic modules or create new ones. When expanding, dual-mode initialization reuses low-level features from the most similar prior module and randomly initializes higher layers for plasticity. TADEN is evaluated on MiniHack and Atari environments, outperforming major baselines like EWC, CLEAR, P&C, MoE, and SANE.

**Strengths:**

* The task-aware expansion using Wasserstein distance between task-specific feature representations is a strong and well-motivated contribution.
* Across both MiniHack and Atari benchmarks, TADEN achieves top performance in average reward and forgetting mitigation with fewer parameters than other architecture-based CRL methods.
* The paper is well-organized, logically progressing from motivation to design and experimentation.

**Weaknesses:**

* While the paper mentions dynamic expansion, the number of modules still grows with task diversity. This may become infeasible for long task sequences.
* Using Wasserstein distance on visual embeddings may not generalize across tasks with abstract (non-visual) differences. Some theoretical justification or sensitivity analysis is missing.
* Small spelling mistakes e.g. in Chapter 5 first sentence:  "DATEN"
* Although major CRL methods are included, recent large-scale continual RL frameworks (e.g., Continual-World benchmarks) are absent. (not required, but maybe why a sentance or two on why it was not considered)

**Questions:**

Your method dynamically expands actor-critic modules based on task similarity, but as task diversity grows, the number of modules, and thus memory and compute, may become large.

Q: Have you considered mechanisms to constrain or merge modules over time?

Q: Do you have ideas for environments that require a very large task diversity?

Q: How robust are Wasserstein distence metrics to non-visual or partially observable tasks, and have you compared it with alternative similarity measures (e.g. Feature-Clustering)

---

### Note · Authors · 2026-01-13

I have read and agree with the venue's withdrawal policy on behalf of myself and my co-authors.